# Migration Testing of GPPS and HIPS Polymers: Swelling Effect Caused by Food Simulants Compared to Real Foods

**DOI:** 10.3390/molecules27030823

**Published:** 2022-01-26

**Authors:** Valeria Guazzotti, Anita Gruner, Mladen Juric, Veronika Hendrich, Angela Störmer, Frank Welle

**Affiliations:** Fraunhofer Institute for Process Engineering and Packaging, IVV. Giggenhauser Straße 35, 85354 Freising, Germany; anita.gruner@ivv.fraunhofer.de (A.G.); mladen.juric@ivv.fraunhofer.de (M.J.); veronika.hendrich@ivv.fraunhofer.de (V.H.); angela.stoermer@ivv.fraunhofer.de (A.S.); frank.welle@ivv.fraunhofer.de (F.W.)

**Keywords:** polystyrene, GPPS = general purpose polystyrene, HIPS = high impact polystyrene, food contact materials, food simulants, migration testing, migration kinetics, polymer swelling, food contact regulation

## Abstract

Migration kinetic data from general purpose polystyrene (GPPS) and high impact polystyrene (HIPS) were generated for a set of model substances as well as styrene monomer and oligomers at different temperatures (20 °C, 40 °C, 60 °C) using food simulants stipulated in the European Regulation (EU) 10/2011 and real foods like milk, cream and olive oil (20 °C, 40 °C). The extent of polymer swelling was characterized gravimetrically and visual changes of the test specimens after migration contact were recorded. Isooctane and 95% ethanol caused strong swelling and visual changes of HIPS, overestimating real migration into foods especially at high temperatures; GPPS was affected by isooctane only at 60 °C. With 50% ethanol, after 10 days contact at 60 °C or 40 °C both polymers were slightly swollen. Contrary, most of the real foods analyzed caused no detectable swelling or visual changes of the investigated polymers. This study demonstrates that the recommendations provided by EU regulations are not always in agreement with the physicochemical properties of styrenic polymers. The critical point remains the selection of adequate food simulants/testing conditions, since the high overestimation of aggressive media can lead to non-compliance of polystyrene materials even if the migration into real food would be of no concern.

## 1. Introduction

Polystyrene homo-polymer, also known commercially as crystal polystyrene or general purpose polystyrene (GPPS), is an amorphous, clear (naturally transparent), hard and rather brittle synthetic polymer made from styrene monomer. High Impact Polystyrene (HIPS) is produced by polymerization of styrene in the presence of a polybutadiene rubber impact modifier (between 5% and 12%). Compared to GPPS, HIPS is an opaque or translucent rather than transparent polymer, with superior impact strength, improved resistance to stress cracking and to crazing caused by organic liquids, oils and fats. HIPS is used on its own or blended with GPPS in specific blend ratios depending on the final functional application requirements [1]. Solid polystyrene can be technologically processed through injection molding, vacuum forming, sheet extrusion or thermoforming, while expanded polystyrene is either extruded or molded in a special process. A wide range of products made of polystyrene are used in several applications such as packaging, building & constructions, medical devices, electronics and consumer goods & appliances (including refrigerators). The sub-segment food packaging in particular is the most dominant application of polystyrene [2]. GPPS is mainly used for the production of transparent containers for a variety of foods and of disposable cups for beverages. HIPS and GPPS-HIPS blends are used in the form of pots e.g., for dairy products, such as yoghurts, as vending cups for beverages such as coffee, tea, chocolate and soup, as containers for ice creams as well as disposable cutlery and utensils Foamed polystyrene trays or containers are used in contact with fast foods, meat, fish, poultry, soft fruit and eggs.

In the Europe Union (EU), the Framework Regulation (EC) 1935/2004 sets out the general principles of safety and inertness for all Food Contact Materials (FCMs). In addition to the general legislation, materials and articles made of plastics intended for food contact applications must comply with the requirements of the Regulation (EU) 10/2011, which establishes a Union List of substances that are permitted for use in the manufacture of plastic FCMs, specifies restrictions and lays down detailed rules for the migration testing conditions to be applied to determine compliance. According to the Regulation (EU) 10/2011, migration experiments shall be performed taking into account the most severe contact conditions of the material with food under foreseeable conditions of use (time/temperature conditions). The representative conditions shall be taken from Annex III (selection of proper food simulant) and Annex V (time/temperature conditions) of the Regulation. The listed food simulants are representative for a food category: acetic acid 3% (*w*/*v*) is assigned for acidic foods, Tenax^®^ for dry food, 10% and 20% (*v*/*v*) ethanol–water mixtures for aqueous and aqueous-alcoholic food, 50% (*v*/*v*) ethanol for dairy products and vegetable oil (e.g., olive oil) for fatty food. Substitute simulants such as 95% (*v*/*v*) ethanol and isooctane can be used to replace olive oil as fatty food simulants in case testing with olive oil is not technically feasible or to screen for migration.

Depending on polarity of polymer, simulant and contact conditions, non-polar polymers are swollen by non-polar solvents and polar polymers by polar solvents. This principle was used [3] for the rapid extraction tests as alternative migration testing and implemented for overall migration testing in the standard EN 1186-15. Polyolefins are swollen by isooctane and polar polymers like polyesters or polyamides by 95% ethanol. These polymers are not or only little swollen by simulants of the reverse polarity [4,5,6]. Styrenic polymers and co-polymers are medium polar. They have strong interactions with both, isooctane and 95 % ethanol. This was recently shown for ABS (acrylonitril-butadiene-styrene-copolymer) [7].

In the draft document “Technical guidelines for compliance testing” [8], milder testing conditions (as normally foreseen with olive oil) for the alternative simulants 95% ethanol and isooctane are listed both for polar and non-polar polymers (including polystyrene) to account for the swelling of polymers in contact with solvent, which may occur and thus accelerate migration. Although not finally published (yet), this guideline represents the current state of scientific knowledge for migration testing. Verification of compliance with a specific migration limit for a material or article can be demonstrated using food in a migration test rather than a food simulant (Regulation EU No. 10/2011 Art.18.6). The results of specific migration testing obtained in food shall prevail over the results obtained in food simulant. However, the use of foods for specific migration testing of materials and articles not yet in contact with food, may pose practical constraints, i.e., food composition change during testing, and analytical difficulties. Nevertheless, it is recommended to compare migration into simulants with that into real foods in order to check the appropriateness of simulation especially when doubts occur. Such studies exist e.g., for PET [9], melamine formaldehyde resins [10], can coatings [11] or from various materials into dry foods [12].

The substances which can migrate from polystyrene (PS) to foods and beverages are residual monomers (i.e., styrene and butadiene), oligomers (principally styrene dimers and trimers) and any applied additives (e.g., antioxidants, antistatics and extenders). As regard styrene, it is an authorized monomer, listed without a specific restriction. Recently, the European Food Safety Authority (EFSA) was requested by the European Commission to evaluate the safety of styrene for use in plastic food contact materials following the classification by the International Agency for Research on Cancer (IARC) as ‘probably carcinogenic to humans’ [13]. In a first scientific opinion [14] it was concluded that the IARC conclusions mainly obtained from high dose occupational exposure and animal studies by inhalation cannot be directly applied to the assessment of consumer risk by oral exposure. Genotoxicity associated with oral exposure cannot be excluded but a systematic review of genotoxicity and mechanistic data, comparative toxicokinetics and analysis of species differences is further required for assessing the safety of styrene and styrene oligomers presence in FCM.

The migration of styrene from PS packaging into food has been extensively studied in the last decades in various foods and simulants. Styrene migration strongly depends on its residual level content in the polymer, the fat content of the food as well as the storage temperature and time conditions [15,16,17,18,19,20,21,22,23]. An evaluation of the migration of styrene monomer and oligomers to foods and in their respective PS packagings in comparison with previous data was carried out in 2014 [24]. The concentrations in various foods were found comparable with literature data reported since the 1980s. According to this review, the level of styrene in foods in contact with PS packaging range from 2.6 to 163 µg/kg (ppb). Styrene dimers and trimers are present in higher concentrations in PS food contact materials than the monomer however; their migration to food is limited, ranging from 2 to 4.8 µg/kg, because of their high partitioning coefficients *K* values and their low diffusion coefficients *D* in PS products.

Controversial data are reported concerning the migration levels of styrene monomer and oligomers into various food simulants/solvents and on the comparison of migration into real food with the corresponding authorized food simulant (especially the alternative simulants). Styrene migration in 3.5% fat milk was assumed to be comparable to that of the 50% ethanol [25], in another study to that of 15% ethanol [20]. Migration testing employing aqueous food simulants lead to under-estimation of the migration processes occurring in real food products [26,27,28,29,30]. On the other hand, it has been recognized that certain food simulating liquids like ethanol/water mixtures, n-heptane or isooctane strongly interact with styrene based materials leading to an overestimation of the migration which results into too high diffusion coefficients due to swelling and stress cracks [27,31,32].

The aim of this study was to systematically investigate the migration behavior from GPPS and HIPS polymers and simultaneously the swelling effect using both, food simulants stipulated in EU regulations and real foods. Migration kinetic data from extruded sheets were obtained for a set of spiked model substances as well as for the monomer styrene and styrene oligomers at different time/temperature conditions in 50% ethanol, isooctane and 95% ethanol. Migration of the volatile compounds were quantified in UHT milk (3.5% fat), UHT cream (30% fat) and olive oil. The extent of polymer swelling was characterized gravimetrically and visual changes of the test specimens were recorded after migration contact at temperatures of 20 °C, 40 °C and 60 °C with following food simulants: 3% acetic acid, 10% ethanol, 50% ethanol, 95% ethanol and isooctane. Additionally, several food types (lard, butter, fish oil, orange juice, coffee beans, coffee, noodles, oat flakes, wheat loops and water) as well as the alternative fatty food simulant Miglyol^®^ 812 were gravimetrically investigated for their interactions with GPPS and HIPS under different storage conditions. They were selected in order to cover the majority of foreseeable food contact applications for these materials.

Obtained data show that the use of isooctane as well of high concentrations of ethanol-water, especially at high temperatures, causes swelling and physical changes of polystyrene and results in significantly higher migration values compared to the investigated foods, which tested under typical contact condition do not show polymer interactions.

It is expected that the results of this study will assist in acquiring information about the selection of suitable migration testing conditions for styrenic polymers, especially if a migration limit for styrene and styrene oligomers will be set out in the future, considering the swelling effect caused by some food simulants prescribed in EU regulations.

## 2. Results

### 2.1. Spiking Levels of Model Substances, Residual Content of Styrene Monomer and Styrene Oligomers

The samples used for the migration kinetics studies were artificially spiked with model substances during extrusion. The initial concentration (*C_p_*_,0_) of the spiked substances as well as the residual content of styrene monomer and styrene oligomers in the investigated polystyrene sheets (GPPS and HIPS) are summarized in Table 1. Concerning the model substances introduced during extrusion, their concentration range was found suitable to get migration values above the detection limits of the applied analytical methods. Additionally low standard deviations were observed, indicating sufficiently homogeneous distributions in the sample sheets. A single extraction was sufficient to yield more than 90% amount of the investigated substances. Calculated recoveries were determined between 75% and 125%.

### 2.2. Migration, Weight Increase and Visual Changes of GPPS and HIPS with Food Simulants

The migration and weight increase from GPPS and HIPS into food simulants at the tested temperatures are depicted in Table 1 to Figure 11. All experiments were carried out in triplicate and the results were averaged. Tabulated data (mean values and standard deviation of each measurement) are reported in the Appendix A. Line graphs of the depicted migration curves represent the connection of the relative migration (dependent variable) over time (independent variable). No fitting of the data was done. The relative migration represents the percentage of substance that migrates from polymer to food simulant. It was calculated by dividing the mass of each substance recovered in the food simulant by the initial mass of the same substance determined in the polymer sheets at the beginning of the study (*C_p_*_,0_ given in Table 1). This way, it is possible to compare the two investigated polymers and appreciate the physical-chemical migration behavior for each particular compound considered.

#### 2.2.1. Migration Kinetics and Weight Increase Using the Food Simulant Isooctane

The migration kinetics and weight increase of GPPS and HIPS into isooctane at 60 °C are reported in Figure 1.

At this temperature, the relative migration values of the model substances and of styrene monomer and oligomers are very high and substantially similar for all the substances regardless of their molecular size, which reflects an extraction from the material rather than a diffusion-controlled migration process. After 10 days of contact it ranges from approx. 40 to 120% in case of GPPS and from approx. 40% to 80% in case of HIPS.

The migration curves for all the investigated substances show a steady increase until the equilibrium is reached after approx. 20 days in case of GPPS and already after approx. 3 days in case of HIPS. The relative migration values higher than 100% in GPPS are most probably due to the analytical uncertainties of the extraction and migration values as well as to possible heterogeneity of the initial concentrations (see reported SD in Table 1). Styrene and the volatile test substances are completely transferred from the 350 µm GPPS sheets into isooctane during the observation period.

The gravimetric experiments show a significant weight increase of GPPS reaching 75% after 50 days. A slightly lower, but still clear increase was found on HIPS reaching 40% after 50 days. It is worth to be noted that at the testing conditions 60 °C for 10 days and 60 °C for 1 day using isooctane as food simulant, both investigated polymers are significantly swollen. This is demonstrated with a measured weight increase of 45% and 12% for GPPS and 30% and 21% for HIPS, respectively.

Additionally, during the kinetic migration experiments in isooctane, visual changes of the GPPS and HIPS test specimens were monitored. The HIPS materials were deformed (rolled up) already after 4 h contact and become opaque and molded after 5 days; while the GPPS materials got softer and cloudy already after 2 h and became deformed (rolled up) after 8 days (see Appendix A).

The migration kinetics and weight increase of GPPS and HIPS into isooctane at 40 °C are shown in Figure 2.

At 40 °C the relative migration of the model substances and of styrene monomer and oligomers from GPPS in isooctane is drastically lower compared to 60 °C. After 50 days contact it ranges from 4% to 12% for the volatile substances (chlorobenzene and styrene), from 3% to 4% for the non-volatiles (phenyl cyclohexane, benzophenone, methyl stearate) and from 4% to 10% for the oligomers. The migration values are still increasing without approaching equilibrium. In case of HIPS, migration values in isooctane at 40 °C are similar or only slightly lower compared to the one obtained at 60 °C. The relative migration values range from 60% to 90% for the volatiles and 30% to 65% for the low volatiles (including the oligomers) after 10 days. The migration curves obtained for the volatile substances reach equilibrium already after approx. 1 day, while for the other compounds it takes approx. 14 days to reach the equilibrium. Interestingly, if compared with 60 °C, the weight increase measured for GPPS at 40 °C was drastically lower. It reached only 1% after 50 days and after 10 days 0.2%. Also in HIPS the measured weight increase was lower compared to 60 °C. It follows a steady increase, reaching up to 13% after 16 days and afterwards approached a plateau with 14% weight increase after 50 days. It is worth to be noted that at the testing condition 40 °C for 10 days using isooctane as food simulant, the HIPS polymer is significantly swollen, which is demonstrated with a measured weight increase of the test specimen of 12%.

Visual changes of the GPPS polymer were observed only after 17 days (softer, cloudy) while for the HIPS specimen, changes were observed already after 4 h contact with a color change from transparent to white.

The migration kinetics and the weight increase of HIPS into isooctane at 20 °C are shown in Figure 3. The relative migration from GPPS was low (see Appendix A) and reached after 10 days only 1% for the volatile model substances and approx. 0.3% for the higher molecular weight compounds. Migration of oligomers from GPPS into isooctane at 20 °C was not detectable.

At 20 °C in isooctane, the relative migration from HIPS of the volatiles after 10 days was between 55% and 65%. It reached equilibrium after approx. 30 days, while for the other compounds (including the oligomers) it was between 10% and 25% after 10 days but did not approach equilibrium even after 120 days.

For GPPS no weight increase (<0.1%) was measurable and no visual changes were observed even after 73 days of contact with isooctane. For HIPS the weight increased slowly, reaching up to 6% after 120 days. It is worth to be noted that at the testing condition 20 °C for 10 days using isooctane as food simulant, the HIPS polymer is significantly swollen which is demonstrated with a measured weight increase of the test specimen of 3%. Additionally, visual changes of the HIPS specimen were observed already after 2 days contact as HIPS became cloudy (see Appendix A).

#### 2.2.2. Migration Kinetics and Weight Increase Using the Food Simulant 95% Ethanol

The migration kinetics and the weight increase of GPPS and HIPS into 95% ethanol at 60 °C are shown in Figure 4.

The relative migration of the model substances and styrene into 95% ethanol are high. However, for both investigated polymers, significant differences in the migration behavior of the volatile substances compared with the lower volatiles were observed. In case of the volatile substances toluene, chlorobenzene and styrene, the migration curves are steeply rising and reach approx. 80% after 10 days of contact from both GPPS and HIPS approaching plateau after approximately 14 days. The migration curves of the lower volatiles increased more slowly and were still increasing after 115 days contact. Migration was at most 30% for benzophenone after 10 days. Migration of styrene oligomers from GPPS in 95% ethanol was drastically lower compared to isooctane tested at the same temperature. Also in case of HIPS, it was significantly lower. After 100 days oligomer migration reached max. 40% for GPPS and 20% for HIPS.

As regard the weight increase in 95% ethanol at 60 °C, in an initial step (after 1 day contact) the measured weight increase for both polymers reaches 3% and approaches more slowly up to 15% after 200 days in case of GPPS and up to 34% after 273 days for HIPS (see Appendix A). At the testing conditions 60 °C for 10 days or 60 °C for 1 day using 95% ethanol as food simulant, the GPPS and HIPS polymers are significantly swollen. Additionally, the test specimens of both polymers used for the migration experiments were deformed (curved) already after 1 day contact (see Appendix A).

The migration kinetics and weight increase of GPPS and HIPS into 95% ethanol at 40 °C are depicted in Figure 5. The relative migration of oligomers into 95% ethanol at 40 °C was below 1% after 10 days (see Appendix A).

As at 60 °C, the relative migration of the volatiles at 40 °C is higher compared to the lower volatiles, reaching up to 20% (GPPS) and 30% (HIPS) after 10 days. The migration of the other compounds increases slowly and remains below 10% after 100 days.

Regarding the weight increase using 95% ethanol at 40 °C, an initial linear rapid increase was observed. After 10 days the weight increase reached approx. 3% for both poly-mers. However, significant visible changes of the test specimens were not noted. Only the HIPS samples appeared slightly yellow only after 44 days contact with 95% ethanol at 40 °C (see Appendix A).

The migration kinetics and weight increase of GPPS and HIPS into 95% ethanol at 20 °C are depicted in Figure 6. Relative migration of styrene oligomers from GPPS and HIPS in 95% ethanol at 20 °C (see Appendix A) was below 1% until 105 days (GPPS) and 202 days (HIPS).

Migration of the most volatile compounds (toluene, chlorobenzene and styrene) did not flatten off even after 120 days or even 200 days tested for HIPS (data not shown). After 10 days the relative migration of these volatiles did not exceed 5% from both polymers. The migration of the higher molecular weight compounds was at or below detection limit. From the weight increase obtained at 20 °C, it can be noted that also at 20 °C the use of 95% ethanol as food simulant causes measurable weight increase for GPPS and HIPS with 1.3% and 1.7% after 10 days, respectively. Visual changes were not appreciable.

#### 2.2.3. Migration Kinetics and Weight Increase Using the Food Simulant 50% Ethanol

The migration kinetics and the weight increase of GPPS and HIPS into 50% ethanol at 60 °C are shown in Figure 7. Relative migration of styrene oligomers for both polymers in 50% ethanol at 60 °C was below 1% up to 50 days contact (data not shown).

The migration of the target volatile compounds increased over time. After 10 days of contact, migration of volatiles from GPPS reached 13% (styrene) or 27% (toluene and chlorobenzene) and from HIPS 20%. In case of low volatiles compounds migration remained in the range of the detection limits. As regard the weight increase related to 50% ethanol at 60 °C, after 10 days migration contact, a weight increase of 1.2% for GPPS and 1.3% for HIPS was measured. For GPPS it remained constant until the last observed time point (127 days, data not shown), while for HIPS it increased after 50 days approaching plateau at 10% after 200 days (data not shown). No visual changes were observed for GPPS while for HIPS they were appreciable. Already after 5 days contact the HIPS sample materials became cloudy; and after 16 days the color changed to yellow. (see Appendix A).

The migration kinetics of target substances from GPPS and HIPS in 50% ethanol at 40 °C are shown in Figure 8. Migration of oligomers was not detectable.

The migration extent from GPPS was lower than from HIPS. After 10 days, migration of styrene from GPPS was not detectable, the relative migration of toluene reaches 4%, while in case of HIPS the relative migration of the volatiles was approx. 10%. In case of low volatiles compounds migration remained in the range of the detection limits. Migration of styrene from GPPS was detectable only after 20 days and increased reaching 5% only after 50 days whereas at the same condition it reached 15% in case of HIPS. After 10 days migration contact in 50% ethanol at 40 °C a weight increase corresponding to 1.1% for GPPS and 1.3% for HIPS was observed. The weight increase remained constant until the end of the kinetic experiments with 127 days and 159 days for GPPS and HIPS respectively. No visual changes were appreciable using 50% ethanol at 40 °C until 44 days for GPPS, when the samples became slightly cloudy. In case of HIPS, the first visual changes were observed after 64 days, when the samples became slightly yellow.

The migration kinetics from HIPS in 50% ethanol at 20 °C are shown in Figure 9. Relative migration of styrene oligomers from HIPS in 50% ethanol at 20 °C was below 0.5% until 157 days (data not shown). Migration from GPPS was below the analytical detection limits.

The relative migration of styrene and the other volatiles was approx. 2–2.5% after 10 days. In case of low volatiles compounds migration remained in the range of the quantification limits. The maximum weight increase was 1.1% reached after 44 days both for GPPS and HIPS. It is to worth to be noted that at the testing condition 20 °C for 10 days using 50% ethanol as food simulant, the GPPS and HIPS polymer are only slightly swollen (0.7 and 0.9% respectively). No visual changes of the test specimens were observed until the last kinetic time point.

#### 2.2.4. Weight Increase Using the Food Simulants 20% Ethanol, 10% Ethanol and 3% Acetic Acid

The results of the weight increase of GPPS and HIPS in contact with the aqueous food simulants are given in Figure 10 and Figure 11.

In case of GPPS, weight increase was measurable only after contact with 20% ethanol while remained < 0.1% using 10% ethanol and 3% acetic acid. With 20% ethanol it was found constant at 60 °C and it increases at 40 °C until 20 days reaching a weight increase of 0.5%. At 20 °C, the weight of GPPS increases from 0.2% after 10 days to 0.4 after 90 days. Regarding HIPS, a weight increase was measurable at all tested conditions. With 20% and 10% ethanol at 60 °C the weight increase showed two sequential linear regions: in the first part (until approx. 100 days) weight increased slowly remaining < 1% while in the second part it increases faster without reaching a plateau. At the end of the storage time, a weight increase of 3.8% after 200 days for 20% ethanol and 3.3% after 273 days for 10% ethanol was detected. After 10 days contact with the food simulant 20% ethanol, the weight increase of the HIPS polymer was determined to be 0.4% at 20 °C, 0.6% at 40 °C and 0.7% at 60 °C. After 10 days contact with the food simulant 10% ethanol, the weight increase of the HIPS polymer was determined to be 0.25% at 20 °C, 0.3% at 40 °C and 0.4% at 60 °C. In the food simulant 3% acetic acid the higher weight increase was determined after 273 days at 60 °C (1.2%). After a storage time of only 10 days the weight increase was 0.2% at 20 °C and 40 °C, respectively, and 0.4% at 60 °C.

Concerning visual testing after migration experiments, for GPPS no visual changes were observed after contact with the food simulants 20% ethanol, 10% ethanol and 3% acetic acid at 20 °C, 40 °C and 60 °C within the storage of up to 140 days. For HIPS, some visual changes were observed with these simulants only at 40 and 60 °C after prolonged times of exposure. With 20% ethanol the samples changed color to slightly yellow after 16 days at 60 °C and after 126 days at 40 °C. With 10% ethanol and 3% acetic acid samples become slightly yellow after 30 days at 60 °C and after 126 days at 40 °C.

### 2.3. Migration and Weight Increase of GPPS and HIPS in Contact with Real Food

#### 2.3.1. Migration and Weight Increase Using Milk, Cream and Olive Oil

The migration kinetic of the lower molecular weight substances (toluene, chlorobenzene and styrene) was tested from GPPS and HIPS up to a storage time of 60 days into UHT milk with 3.5% fat content, UHT cream with 30% fat content and olive oil at 20 °C. Additionally storage conditions of 10 d at 40 °C were also investigated for the milk, cream and olive oil. At the same conditions, the weight increase of the polymer sheets before and after contact with the foods was measured. In Table 2 (GPPS) and Table 3 (HIPS) the relative migration for toluene, chlorobenzene and styrene and the weight increase after contact with each food is reported. The relative migration represents the percentage of substance that migrates from the polymer to the food. For styrene, the migrated amount into each food (in µg/kg = ppb) is given as well. It was calculated considering the total surface of the specimens and applying the EU cube model: 6 dm^2^/kg.

Migration into food from HIPS was higher compared to GPPS. The lowest migration of styrene was observed in milk after 15 days at 20 °C; it was approx. 90 µg/kg from HIPS, roughly three times higher than from GPPS (approx. 30 µg/kg). For both polymers, migration into milk (3.5% fat) was lower as into cream (with 30% fat content) and olive oil. Migration into cream and olive oil was comparable; the slightly higher values found in cream could be due to the uncertainty of the method, additionally the GPPS material after contact with cream and olive oil broke into pieces with the consequence that the “edge effect” increased. Migration of the investigated substances from GPPS after 10 days at 40 °C into food was higher compared to the condition 60 days at 20 °C. In olive oil and milk, the migration was twice as high compared to 20 °C. Migration of styrene from GPPS ranged from approx. 30 µg/kg (in milk after 15 d at 20 °C) to approx. 220 µg/kg (in cream after 10 d at 40 °C). No significant weight increase (<0.3%) was observed for GPPS at all the tested conditions using UHT milk, UHT cream and olive oil. However, GPPS material after contact with cream and olive oil became significantly brittle and broke into pieces. Therefore, it was not possible to determine precise weight increase of GPPS in these foods. Migration of the investigated substances from HIPS after 10 days at 40 °C into food was comparable to the migration extent at the condition 60 days at 20 °C. In case of styrene it was equal to approx. 150 µg/kg, 270 µg/kg and 220 µg/kg, respectively, in milk, cream and olive oil. These values correspond to a relative migration of approx. 3.6%, 6% and 5%, respectively. No significant weight increase (<0.6%) was observed for HIPS at all the tested conditions using UHT milk, UHT cream and olive oil except at 60 days at 20 °C with olive oil. In this case, a weight increase of 2.3% was determined. No visual changes in appearance of GPPS or HIPS with all the tested foods were observed.

#### 2.3.2. Weight Increase of the Polymers after Contact with Several Foods under Real Application of Use

In order to achieve better understanding of the weight increase on GPPS and HIPS polymers eventually caused by real foods under real application conditions, several additional foods and the synthetic tricglyceride Miglyol^®^812 were investigated at storage at 5 °C resp. 20 °C for 10, 20, 30 and 50 days using the GPPS and HIPS test samples (Table 4).

As expected dry foods, like ground coffee beans, oat flakes or wheat loops do not cause a significant weight increase at all (< 0.1%). In case of contact with fatty food, such as lard, butter or fish oil, a slight weight increase (< 1%) was observed. It should be noted that cleaning the polymer strips after the fatty food contact was more difficult than for the other foods. This could be also a possible reason for the higher recorded weight for the HIPS polymer samples after contact with fatty food. The weight increase recorded after contact with Miglyol^®^812, which is an alternative food simulant for olive oil, was in the range of 4–8%. This weight increase was higher compared to olive oil and in case of HIPS even higher than the alternative food simulants 95% ethanol and isooctane tested under the same conditions at 20 °C. In case of GPPS the weight increase was higher using 95% ethanol and isooctane compared to Miglyol^®^812.

In Table 5 the weight increase (%) of GPPS and HIPS after contact with water at different temperatures as well as with brewed coffee is reported.

Measurements were made during cooling process at ambient temperature (after 10, 20 and 30 min) and, in case of hot coffee, also at constant temperature of 100 °C. Except for brewed coffee (100 °C) that reached maximum weight increase of 1.2% with HIPS, all other results were below 1%.

In contact with hot coffee at constant temperature of 100 °C as well as at filling at 90 °C and subsequent cooling, the weight increase of HIPS was equal to 0.4% after 10 min and further increased. Already after 10 min contact with hot coffee, the HIPS stripe appeared curved (see Appendix A).

The results indicate clearly that nearly all tested foods did not show a significant weight increase of HIPS. All have been below 1%. The only exceptions were Miglyol^®^812 and brewed coffee at 100 °C after 30 min contact.

Visual changes and in appearance were not observed during contact with foods except for Miglyol, where the strips were discolored. In addition, in water and brewed coffee at 100 and 80 °C the strips were slightly bowed (see Appendix A). These findings are in clear opposition with the strong weight increase observed for the simulants isooctane, 95% ethanol and 50% ethanol.

## 3. Discussion

The investigated GPPS and HIPS sheets showed normal levels of residual monomer and oligomers when compared with previous reported values for PS food contact materials and articles [24,32,33,34]. Additionally, the sheets were artificially spiked during extrusion with model compounds that are widely used for migration and functional barrier studies [35,36,37] as well as for so-called challenge tests for determining the cleaning efficiency of recycling processes [38,39,40]. These artificially added substances represent the following general categories: volatile and non-polar (toluene and chlorobenzene), non-volatile and non-polar (phenyl cyclohexane), non-volatile and polar (benzophenone and methyl stearate). The artificially spiked substances were included in the study in order to increase the amount and variety of substances for the migration kinetics as well in order to add polar substances.

Isooctane is known to induce the swelling of non-polar and medium polar polymers like polystyrene according to the principle “like dissolves like” and consequently, to modify the transport properties of the material [41]. The results of this study show that contact with isooctane caused a clear weight increase (i.e., uptake of isooctane) of the GPPS and HIPS polymers, which is most probably due to this swelling effect. At 60 °C the polymers are completely swollen with a weight increase after 10 days of 45% for GPPS and 30% for HIPS, respectively. The swelling affects the migration kinetics. At 60 °C testing temperature, the initial slope (only visually appreciated) of the migration curves (cf. Figure 1) in isooctane was very high and substantially similar for all the substances, regardless of their molecular size and polarity. From the diffusion theory [42], the level of migration from polymer material depends on the initial concentration (*C_p_*_,0_) and nature of the substance with its molecular size and solubility, the type of material and the food or food simulant in contact. In this study, the migration kinetics obtained using isooctane reflect more an extraction from the material rather than a diffusion-controlled migration process.

When testing GPPS and HIPS using isooctane at 40 °C or 20 °C (cf. Figure 2 and Figure 3), differences for these two polymers were observed.

GPPS was not significantly affected at 40 °C and 20 °C by isooctane with a weight increase below 1%. Migration levels from GPPS were significantly lower compared to 60 °C. At 40 °C migration of the larger components especially of the oligomers from GPPS does not follow diffusion theory. The percentage of migration of the dimers (MW 208 g/mol) is not higher than that of the trimers (MW 316 g/mol), but in a similar range. This suggests a swelling in the first micrometers of the test sheets.

In contrast to GPPS at these lower temperatures, the HIPS polymer was still strongly swollen with a weight increase of 12% and 3% respectively after 10 days at 40 °C and 10 days at 20 °C. Correspondingly, migration values in isooctane at 40 °C are similar or only slightly lower compared to the one obtained at 60 °C. Only when testing HIPS at 20 °C some differences in the migration behavior according to the nature of the substances (as expected from diffusion theory) were noted. These results are in agreement with previous studies [43], which indicate that the rubber content (styrene-butadiene-copolymer) in HIPS can be an influencing factor for migration of chemicals such as styrene. At 60 °C in isooctane HIPS was less swollen than GPPS, which can be an effect of its increased heat resistance. However, at lower temperatures HIPS was more swollen than GPPS, possibly because of the interactions of isooctane with the rubber. The strong swelling of GPPS at 60 °C related to the lower temperatures and related to HIPS is explained by solvent crazing and stress cracking [44].

Looking at the migration and weight increase using 95% ethanol, a similar behavior was observed for GPPS and HIPS. 95% ethanol is more polar compared to isooctane, but still causes weight increase (and therefore swelling) of both polymers at all tested temperatures. At all tested temperatures relative migration values are high but depending on the nature of the substances. The time dependent migration of the most volatiles (toluene, chlorobenzene, styrene) was bigger compared to the higher molecular weight test substances (phenyl cyclohexane, benzophenone and methyl stearate with 160, 182 and 298 g/mol respectively).

According to the Regulation (EU) 10/2011, 50% ethanol is assigned to the food category dairy products. The EU 10/2011 test scheme foresees for specific migration, accelerated test conditions at elevated temperature. For example, testing for 10 days at 60 °C simulate contact times longer than 30 days at room temperature and below, while testing for 10 days at 40 °C shall cover all storage times at refrigerated and frozen conditions including hot-fill conditions and testing for 10 days at 20 °C all storage times at frozen condition. Using 50% ethanol, only little weight increase at 60 °C, 40 °C and 20 °C was found for both the polymers investigated in the present study.

Contrary, none of the tested foods did cause a significant weight increase of GPPS or HIPS material or did induce visual changes under real conditions of usage.

Alternative testing conditions (milder compared to the conventional) for the simulants 95% ethanol and isooctane depending on the polarity of the polymer are proposed in the draft document “Technical guidelines for compliance testing” [8] in consideration of the swelling effect or as substitute test when oil testing is not feasible. In these guidelines, the polymers are divided into two groups, those that contain carbon and hydrogen only and those that contain additionally other atoms. Both polystyrenes, GPPS and HIPS belong to the first group. According [8], migration testing for the long-term storage condition 10 days/60 °C shall be carried out with 95% ethanol under the same time-temperature conditions as oil, while with isooctane at 10 days/20 °C. For screening purposes, testing conditions are specified for the polystyrene polymer: 10 days at 60 °C, 40 °C and 20 °C in oil are reduced to 1 day for both 95 % ethanol and isooctane, except for 95 % ethanol at 20 °C, which shall be tested with 10 days contact.

In Table 6 migration of styrene in food simulants and real foods at different time/temperature testing conditions, swelling (as % weight increase) and visual changes of the sample material are reported. For the food simulants, not always exactly 10 days were measured; data were in this case extrapolated. For the alternative fatty food simulants (95% ethanol and isooctane) official and alternative testing conditions are summarized. The alternative testing conditions are in accordance with the draft guideline [8] (page 59 Table 7: “Test conditions for screening food simulants compared to vegetable oil for specific migration”).

The alternative fatty food simulants isooctane and 95% ethanol cause significant swelling of the HIPS polymer (both considering official and alternative testing condition), which is further increased by higher contact temperatures. The effect is that obtained migration values tend to overestimate the actual migration into real foods. In real foods even after 10 days contact at 40 °C the weight increase remained below 1%.

In case of HIPS, isooctane overestimate real migration into foods at all temperatures.

Concerning GPPS, the migration levels as well as the swelling effect of isooctane strongly depend on the temperature. At 60 °C styrene migration from GPPS reaches approx. 4000 µg/kg after 10 days and the corresponding weight increase is 45%. GPPS exposed to temperatures of 50 °C is affected by stress-cracking [23], which lead to an overestimation of the migration under the actual condition of use. The weight increase of GPPS after contact with isooctane at 20 °C and 40 °C was below 1%; migration of styrene after 1 day and 10 days at 40 °C underestimates levels found into milk, cream and olive oil tested at 10 days 40 °C.

Testing both HIPS and GPPS with 95% ethanol at the condition 10 days at 40 °C largely overestimates the real migration into milk, cream and olive oil. While testing with 95% ethanol at the condition 1 day at 40 °C styrene migration fits to the real foods.

With 50% ethanol, after 10 days contact at 60 °C or 40 °C both the GPPS and HIPS polymers are slightly swollen (weight increase < 1.1%). Contrary, after 60 days contact with milk, cream and oil at 20 °C or after 10 days 40 °C, the weight increase was below 0.4%. The migration values in this food simulant at 10 days/60 °C are higher (especially in case of GPPS) compared to the real food tested at 10 days/40 °C. Testing both polymers with 50% ethanol at the condition 10 days/20 °C underestimates the real migration into foods. In case of HIPS, migration levels at 10 days/40 °C fit the styrene migration at the same condition in cream and olive oil but slightly overestimate that in milk. For GPPS, although a measurable weight increase was observed at the condition 10 days/40°C, migration of styrene remained below the detection limit.

In conclusion, contact conditions for migration testing of 10 days at 60 °C using isooctane, 95 % ethanol and 50 % ethanol are too severe for both GPPS and HIPS. Other combination of time/temperature conditions should be applied. These findings are in good agreement with previously reported results for other polymers, such as for acrylonitrile-butadiene-styrene copolymer (ABS) [7], styrene-acrylonitrile copolymer (SAN) [45] and polyethylene terephthalate (PET) [46].

Migration of styrene from HIPS tested using migration cells at 10 days/40 °C into foods with 30% fat content was reported by Linssen and Reitsma [47] in the range of 240–320 µg/kg; while the levels found in corn oil were higher reaching approx. 400 µg/kg.

It is worth to be noted that the testing conditions applied in this study for the food milk cream and olive oil do not represent a common real application of FCM made of polystyrene and have to be considered a worst-case condition. The majority of the products packed in PS, which are mostly a blend of GPPS and HIPS, are refrigerated (5 °C) with a shelf life of approx. 40 days, (e.g., yoghurt pot). Typical values found in recent food surveys [22,24] report styrene levels into foods taken from the manufacturer or from the retail ranging from 2.6 to 163 µg/kg (ppb) depending on the fat content and the time/temperature storage conditions.

The migration values determined within the present research work for styrene in the real foods are not representative for the levels found in real sample from the market. Additionally, the test stripes were totally immersed in the foods and the simulants. Thus, edge effects occur which lead to an overestimation of the migration under the actual condition of use where foods are packed in containers. However, the experiments for this study were carried out in the same way for food simulants and foods therefore allowing comparisons, which was the intention.

As swelling of GPPS and HIPS was little or not detectable in contact with real foods under realistic application conditions, this must have consequences for the migration testing protocol. According to the Regulation (EU) 10/2011, the use of 95% (*v*/*v*) ethanol and isooctane as simulants for compliance testing of FCM is accepted to screen for specific migration or to replace olive oil without any specification for which polymers this approach is useful. Migration testing with 95 % ethanol and isooctane is a widely used practice to simplify and reduce the time-consuming and expensive analytical procedures. On the other hand, alternative test simulants in order to screen for migration are suitable to show conformity only, but not non-conformity of a plastic food contact material. Furthermore, according to Regulation (EU) 10/2011, Chapter 2 paragraph 2.1.3 (i): in case of physical changes which do not occur under real use, “the migration tests shall be carried out under the worst foreseeable conditions of use in which physical or other changes do not take place”. Such physical changes in GPPS and HIPS were clearly found in this study.

The comparison with foods was done only for the volatile components. The results discussed for styrene were similar to those for the added test substances toluene and chlorobenzene. All three substances show similar relative migrations in the respective foods and simulants. For larger molecules like the oligomers, the swelling might have a stronger effect on the migration. Further research work is envisaged in order to better evaluate the overestimation of the laboratory simulation compared to real food contact applications and to identify the most suitable test conditions for a realistic simulation of the migration.

## 4. Materials and Methods

### 4.1. Sample Materials

The GPPS and HIPS grades used in the present study are standard industrial materials used in the European market for food contact applications. The used GPPS grade has a molecular weight of around 240.000 g/mol, melt flow index of 12 cm^3^/10 min at 200 °C/5 kg load, glass transition temperature of 100 °C and density of 1040 kg/m^3^. The used HIPS grade has a molecular weight of around 190.000 g/mol, melt flow index of 4 cm^3^/10 min at 200 °C/5 kg load, glass transition temperature of 100 °C and density of 1030 kg/m^3^. The pellets (gently provided by the Styrenics Steering Committee of Plastics Europe) were processed using a Collin Lab & Pilot Solutions co-extruder (Maitenbeth, Germany) to produce sheets with a thickness of approx. 350 µm. For both polymers, the temperature in the extruder was approx. 190 °C; the screw speed was approx. 100 rpm.

A mixture of five model substances, with a variety of chemical and physical properties, were spiked homogeneously during polystyrene sheets production as already described in [7]. They were toluene (CAS No. 108-88-3), chlorobenzene (CAS No. 108-90-7), phenyl cyclohexane (CAS No. 771-98-2), benzophenone (CAS No. 119-61-9) and methyl stearate (CAS No. 112-61-8). All chemicals used in the experiments were of analytical grade: toluene, styrene and benzophenone were purchased from Sigma-Aldrich (St. Louis,MO, USA); chlorobenzene from Merck (Darmstadt, Germany); phenyl cyclohexane from Alfa Aesar (Ward Hill, USA) and methyl stearate from Fluka (Seelze, Germany).

### 4.2. Determination of the Initial Concentration (C_p,0_) of Spiked Substances, Styrene Monomer and Styrene Oligomers

The initial concentrations of the spiked substances as well as of the styrene monomer and styrene oligomers were determined in the GPPS and HIPS spiked sheets by extraction with acetone (purchased from Chemsolute, Th.Geyer GmbH, Renningen, Germany). 1.0 g of the material was extracted with 10 mL acetone and stored at 60 °C for 3 days. Triplicate experiments were performed. A subsequent extraction (under the same condition) and recovery experiments were performed to check for exhaustiveness.For recovery experiments, reference sheets (not spiked during extrusion) were used. They were fortified at three levels with the model substances and available commercial standards for styrene dimers (PSS-266) and trimers (PSS-370) purchased from PSS Polymer Standards Service (Mainz, Germany), directly in the vial, extracted and analyzed as the samples.

The extracts were analyzed by gas chromatography with flame ionization detection (GC-FID). The Agilent 6890 GC/FID system (Agilent Technologies, Santa Clara, CA, USA) was used with the following conditions: column: DB 1 (length 20 m, inner diameter 0.18 mm, film thickness 0.18 µm); GC temperature program: 50 °C (2 min), followed by heating at 10 °C min^−1^ to 340 °C (15 min); pre-pressure: 50 kPa hydrogen; split: 10 mL min^−1^. The injection volume was 5 µL with a split ratio of 1:20. An internal standard solution of di-(tert-butyl)-hydroxyanisol (BHA purchased from Fluka, Seelze, Germany) and Tinuvin 234 (Ciba, Basel, Switzerland) was used for all GC analyses to check the stability of the retention times and the performance of the GC analysis.

Quantification of spiked substances and monomer styrene was achieved through calibration with respective standard substances, while styrene oligomers were semi-quantified using the response factor of BHA.

### 4.3. Migration Kinetics in Food Simulants

The migration kinetics of the artificially spiked substances as well as of the styrene monomer and oligomers were determined into the following food simulants: 50% ethanol, 95% ethanol and isooctane, at temperatures of 20 °C, 40 °C and 60 °C, monitoring several time points (from 0.5 h to 273 days, sampling times were adapted depending on the food simulant and contact condition). All solvents used in the experiments were of analytical grade and were purchased from Chemsolute, Th.Geyer GmbH, Renningen, Germany.

Migration contact with the food simulants was performed by total immersion in capped migration vials in triplicate. The used surface/volume ratio was 1 dm^2^ (total surface of the test specimens) per 100 mL simulant. In parallel to the samples, simulant blanks were prepared. At each kinetic time point a 0.5 mL aliquot was taken for analysis.

Quantification of migration into simulants was obtained using gas chromatography coupled with flame ionization detection (GC/FID) by external calibration with standard dilution series in the respective simulants. The same chromatographic conditions as for the quantification of initial concentrations in the polymer were used. To check stability and possible loss by evaporation of the volatile substances, control tests were performed as well. The sensitivity of the analysis for each substance in the respective food simulant was determined on a 95% confidence interval according to the calibration curve method (see Table 7).

### 4.4. Migration of Volatile Substances in Real Foods

The migration kinetics of the volatile substances (toluene, chlorobenzene and styrene) in real foods was determined at 20 °C after 15, 20, 30, 40 and 60 days, additionally the time/temperature condition 10 days/40 °C was investigated. UHT milk (3.5% fat), UHT cream (30% fat) and olive oil were selected as real food. They were not in previous contact with polystyrene and were bought from a local market in Munich, Germany.

The migration tests in real food samples were performed by total immersion, in triplicate. The used surface/volume ratio was 0.2 dm^2^ (total surface) per 20 g (food) and therefore equivalent to 1 dm^2^/100 mL as used for the food simulants. A spatula tip of sodium azide was added to avoid spoilage. After migration contact, 5 g of each food sample (after homogenization) was placed into a headspace vial and analyzed by headspace gas chromatography with flame ionization detection (Headspace-GC-FID).

The Perkin Elmer AutoSystem XL (Perkin Elmer Industries, Shelton, USA) was used, with the following conditions: headspace auto-sampler: Perkin Elmer HS 40 XL, oven temperature: 80 °C or 120 °C (only for olive oil), needle temperature: 110 °C or 130 °C (only for olive oil), transfer line temperature: 120 °C or 140 °C (only for olive oil), equilibration time: 1 h, pressurizing time: 3 min, injection time: 0.05 min, withdrawal time: 1 min; GC column: DB 624 (length 60 m, inner diameter 0.32 mm, film thickness 1.8 µm); GC temperature program: 40 °C (6 min), heating rate 5 °C min^−1^ up to 90 °C and then 10 °C min^−1^ up to 260 °C (10 min), pressure: 120 kPa helium, split: 40 mL min^−1^.

For quantification, external calibration on each food matrix was carried out individually. Standard solutions (4 levels, diluted from a stock solution prepared in ethanol) were added to each foodstuff in twice, after homogenization 5 g of each spiked food level was analyzed to obtain a standard calibration curve. Control standards (known spiked amount of each food) were analyzed in parallel to the samples to check for recovery. The limit of detection of the method was defined as three times the signal-to-noise ratio (see Table 7).

### 4.5. Weight Increase with Food Simulants and Real Foods

The extent of polymer swelling was characterized gravimetrically as weight increase of the test specimens after contact with the simulants or foods. Weight increase for the GPPS and HIPS polymers were obtained after contact with 50% ethanol, 95% ethanol and isooctane at the same time/temperature condition used for the migration experiments. Additionally, 3% acetic acid 10% ethanol and 20% ethanol were investigated for their interactions with GPPS and HIPS at temperatures of 20 °C, 40 °C and 60 °C, adapting sampling times (until max 273 days) on the food simulant and contact condition. As for the migration tests, experiments were performed by total immersion, in triplicate. The used surface/volume ratio was 0.2 dm^2^ (total surface) per 20 mL (simulant).

Weight increase of the investigated polymers in contact with UHT milk, UHT cream and olive oil were obtained simultaneously with the migration experiments (by weighing the test specimens before and after each migration contact preparation). Additionally, other foods (lard, butter, fish oil, orange juice, coffee beans, coffee, noodles, oat flakes, wheat loops and water) as well as the alternative fatty food simulant Miglyol^®^ 812 (CAS No. 73398-61-5: Decanoyl-and Octanoylglycerides) were tested to determine the extent of swelling. The selection of foods and testing conditions was based on the input provided by the Styrenics Steering Committee of Plastics Europe on those foods that are widely in contact with polystyrene in real applications (see Table 4 and Table 5). All investigations were performed in triplicate.

### 4.6. Testing on Visual Changes of the Polymers after Contact with Food Simulants and Real Foods

During the kinetic migration tests and the polymer swelling tests, changes in the appearance of the GPPS and HIPS test specimens were noted and photos were taken at many time intervals. All the photos of the test material taken at the different intervals during the kinetics tests are given in the Appendix A.

## 5. Conclusions

Within this study, migration of model substances, styrene monomer and oligomers from GPPS and HIPS at different temperatures was investigated using food simulants stipulated in EU regulations and real foods (milk, cream and olive oil). At the same, the extent of polymer swelling was characterized gravimetrically after contact with simulants as well as different foods tested under several conditions to cover most of the real applications of these materials. Isooctane and 95% ethanol caused strong swelling and visual changes of GPPS and HIPS, overestimating real migration into foods. Migration and swelling from HIPS was higher compared to GPPS except when testing at 60 °C; thus indicating that the rubber content improved heat resistance but at the same can increase the interactions with non-polar media. With 50% ethanol, after 10 days contact at 60 °C or 40 °C both polymers were slightly swollen. Indeed, none of the tested foods significantly swelled polystyrene under real conditions of usage determined by the lack of weight increase of the polymers. Alternative suitable conditions for testing food contact materials and articles made of polystyrene, in non-swelling conditions, shall be define to mimic at the best way the migration in real food and, in the perspective of a possible restriction for styrene and styrene oligomers, to avoid their “false” non-compliance evaluation.

## Figures and Tables

**Figure 1 molecules-27-00823-f001:**
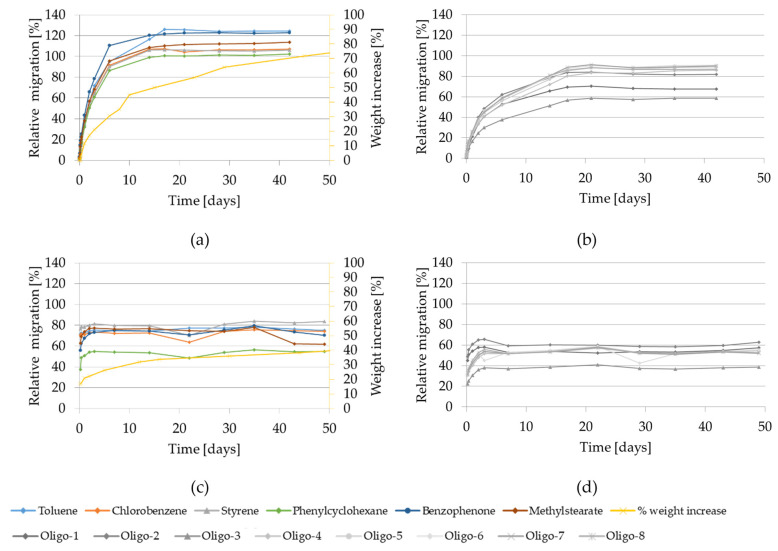
Migration kinetics and weight increase in isooctane at 60 °C: (**a**) Model substances and styrene from GPPS; (**b**) Styrene oligomers from GPPS. (**c**) Model substances and styrene from HIPS; (**d**) Styrene oligomers from HIPS.

**Figure 2 molecules-27-00823-f002:**
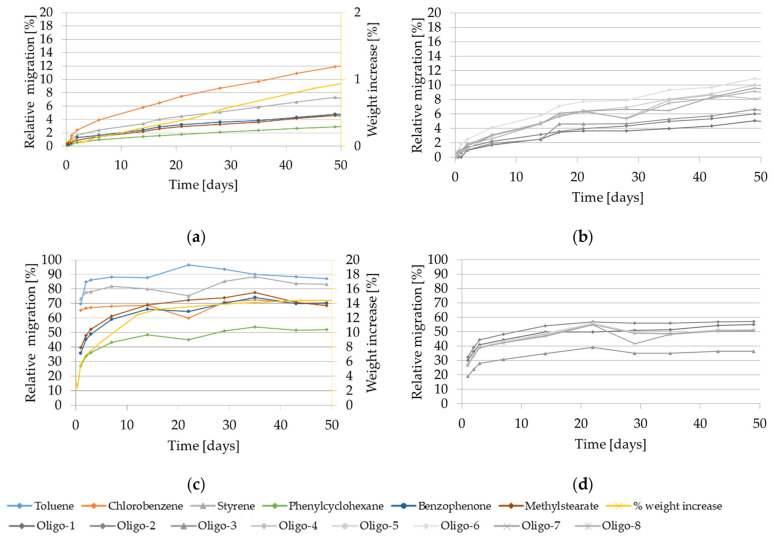
Migration kinetics and weight increase in isooctane at 40 °C: (**a**) Model substances and styrene from GPPS; (**b**) Styrene oligomers from GPPS. (**c**) Model substances and styrene from HIPS; (**d**) Styrene oligomers from HIPS. Note: Due to analytical interferences, migration of toluene from GPPS isooctane at 40 °C could not be measured.

**Figure 3 molecules-27-00823-f003:**
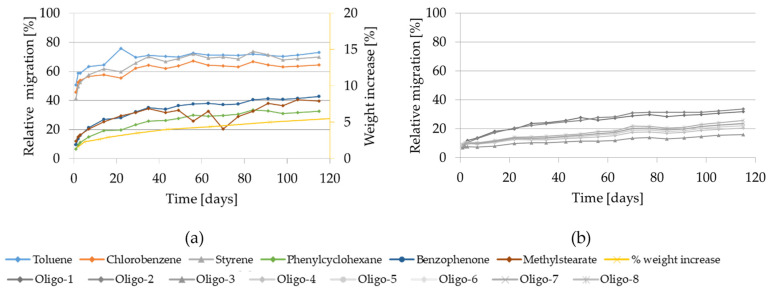
Migration and swelling kinetics in isooctane at 20 °C from HIPS: (**a**) Model substances and styrene; (**b**) Styrene oligomers.

**Figure 4 molecules-27-00823-f004:**
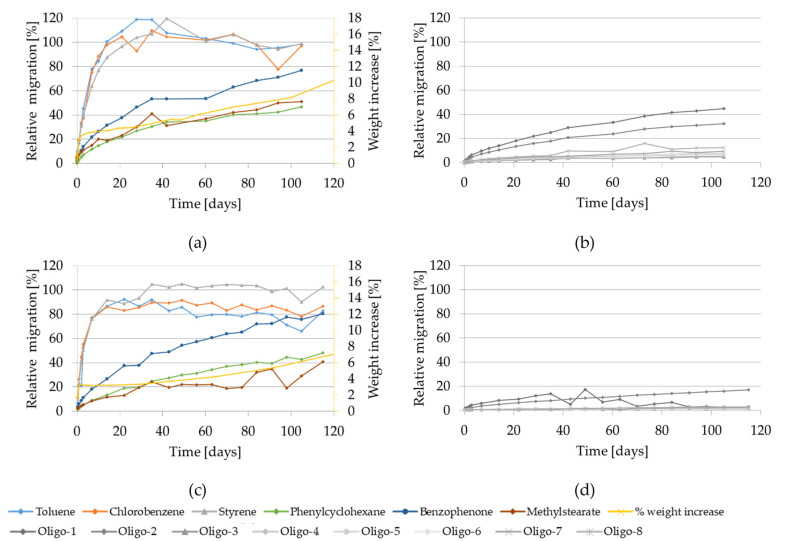
Migration kinetics and weight increase in 95% ethanol at 60 °C: (**a**) Model substances and styrene from GPPS; (**b**) Styrene oligomers from GPPS. (**c**) Model substances and styrene from HIPS; (**d**) Styrene oligomers from HIPS.

**Figure 5 molecules-27-00823-f005:**
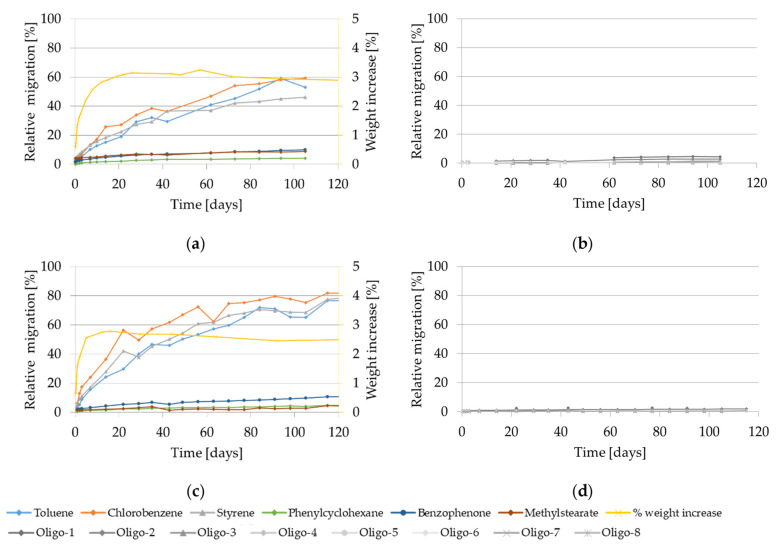
Migration kinetics and weight increase in 95% ethanol at 40 °C: (**a**) Model substances and styrene from GPPS; (**b**) Styrene oligomers from GPPS. (**c**) Model substances and styrene from HIPS; (**d**) Styrene oligomers from HIPS.

**Figure 6 molecules-27-00823-f006:**
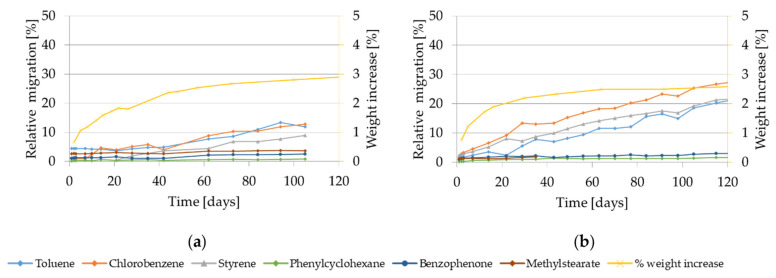
Migration and weight increase in 95% ethanol at 20 °C: (**a**) Model substances and styrene from GPPS; (**b**) Model substances and styrene from HIPS.

**Figure 7 molecules-27-00823-f007:**
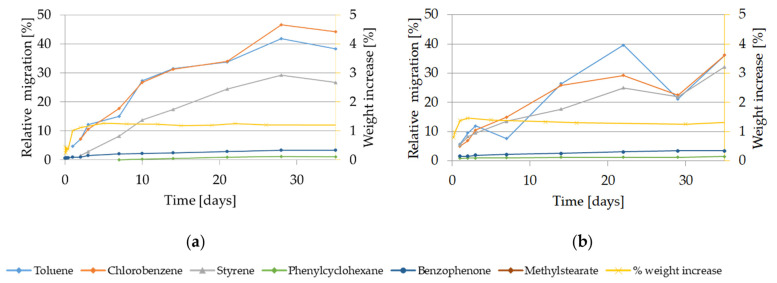
Migration kinetics and weight increase in 50% ethanol at 60 °C: (**a**) Model substances and styrene from GPPS; (**b**) Model substances and styrene from HIPS. Note: Due to analytical interferences, migration of methyl stearate from GPPS and HIPS in HIPS in 50% ethanol at 60 °C can not be measured.

**Figure 8 molecules-27-00823-f008:**
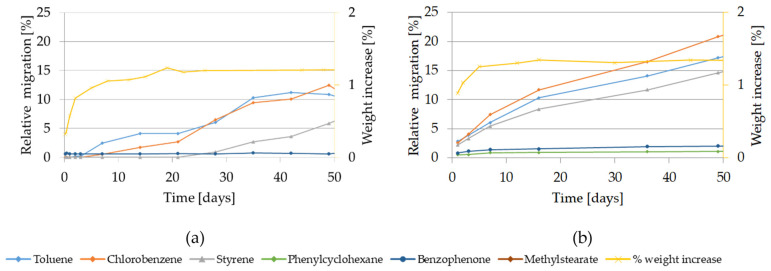
Migration kinetics and weight increase in 50% ethanol at 40 °C: (**a**) Model substances and styrene from GPPS; (**b**) Model substances and styrene from HIPS. Note: Due to analytical interferences, migration of methylstearate from GPPS and HIPS and of phenylcylohexane from GPPS in 50% ethanol at 40 °C can not be measured.

**Figure 9 molecules-27-00823-f009:**
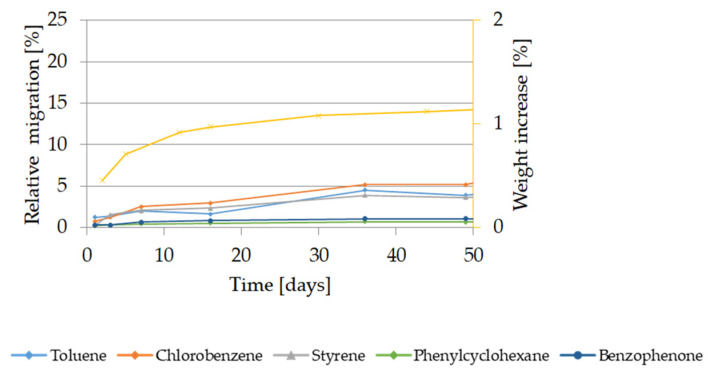
Migration kinetics and weight increase in 50% ethanol at 20 °C from HIPS. Note: Due to analytical interferences, migration of methyl stearate from HIPS in 50% ethanol at 20 °C can not be measured.

**Figure 10 molecules-27-00823-f010:**
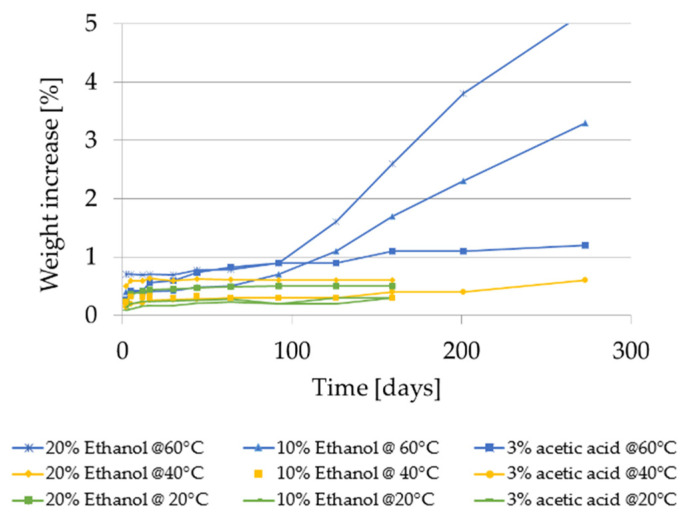
Weight increase of GPPS in 20% ethanol at 20, 40 and 60 °C for storage times up to 140 days.

**Figure 11 molecules-27-00823-f011:**
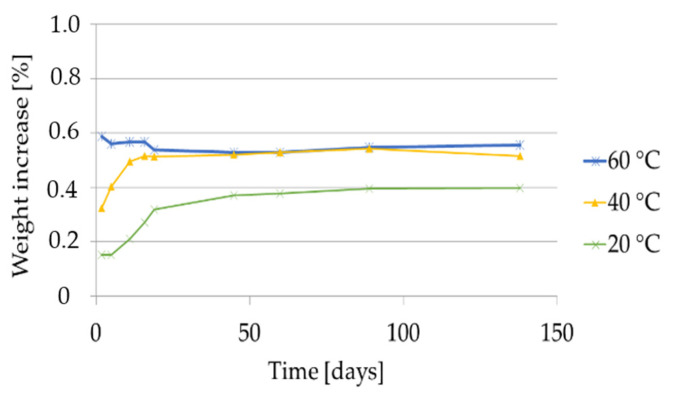
Weight increase of HIPS in aqueous food simulants at 20, 40 and 60 °C for storage times up to 250 days.

**Table 1 molecules-27-00823-t001:** Average initial concentration (*C_p_*_,0_) of the model spiked substances and residual content of styrene monomer and oligomers in the investigated polystyrene (GPPS and HIPS) sheets (±standard deviation of triplicate analysis).

	Average Concentration ± SD [mg/kg]	
Substance	GPPS	HIPS
**Artificially added “spiked” substances**		
Toluene	417 ± 5	763 ± 11
Chlorobenzene	504 ± 5	821 ± 13
Phenyl cyclohexane	705 ± 8	1229 ± 33
Benzophenone	610 ± 5	1002 ± 27
Methyl stearate	718 ± 7	1179 ± 34
**Styrene**	352 ± 3	363 ± 6
**Styrene dimers (sum) ^1^:**	730	1743
1,3-Diphenylpropane: Oligo-1	180 ± 6	134 ± 4
2,4-Di-phenyl-1-butene: Oligo-2	550 ± 7	1609 ± 35
**Styrene trimers (sum) ^1^:**	5807	30290
2,4,6-Triphenyl-1-hexen: Oligo-3	1389 ± 23	4323 ± 152
isomer of 1-Phenyl-4-(1-phenylethyl)-tetraline: Oligo-4	1092 ± 20	6740 ± 230
isomer of 1-Phenyl-4-(1-phenylethyl)-tetraline: Oligo-5	2324 ± 44	13440 ± 455
isomer of 1-Phenyl-4-(1-phenylethyl)-tetraline: Oligo-6	657 ± 14	3684 ± 146
isomer of 1-Phenyl-4-(1-phenylethyl)-tetraline: Oligo-7	260 ± 5	1624 ± 60
isomer of 1-Phenyl-4-(1-phenylethyl)-tetraline: Oligo-8	85 ± 2	479 ± 18

^1^ The styrene oligomers were characterized via their mass spectra and tentatively identified as far as possible by comparison with library spectra and literature data.

**Table 2 molecules-27-00823-t002:** Relative migration [%] of volatiles, migrated amount of styrene in real foods (±standard deviation of triplicate analysis) and weight increase [%] of the polymer at the tested time/temperature conditions from GPPS.

Time/Temperature	Relative Migration [%] into Milk (3.5% Fat)			Migration of Styrene [µg/kg]	Weight Increase [%]
	Toluene	Chlorobenzene	Styrene		
15 d/20 °C	0.4 ± 0.2	1.3 ± 0.2	0.7 ± 0.1	32 ± 5	<0.1
20 d/20 °C	0.5 ± 0.3	1.3 ± 0.3	0.7 ± 0.3	32 ± 12	<0.1
30 d/20 °C	0.6 ± 0.1	1.4 ± 0.2	0.8 ± 0.1	37 ± 6	<0.1
40 d/20 °C	0.7 ± 0.2	1.6 ± 0.3	0.9 ± 0.2	40 ± 8	<0.1
60 d/20 °C	0.8 ± 0.2	1.6 ± 0.2	1.0 ± 0.2	43 ± 9	<0.1
10 d/40 °C	2.8 ± 0.4	3.2 ± 0.2	2.0 ± 0.1	92 ± 3	0.1
	**Relative Migration [%] into Cream (30% Fat)**			**Migration of Styrene [µg/kg]**	**Weight Increase [%] ^1^**
	**Toluene**	**Chlorobenzene**	**Styrene**		
15 d/20 °C	4.0 ± 0.2	3.8 ± 0.2	2.8 ± 0.1	120 ± 4	<0.1
20 d/20 °C	4.2 ± 0.4	4.1 ± 0.4	2.9 ± 0.2	124 ± 9	<0.1
30 d/20 °C	5.2 ± 0.6	5.5 ± 0.4	3.7 ± 0.2	158 ± 8	<0.1
40 d/20 °C	5.5 ± 0.4	5.5 ± 0.3	3.5 ± 0.4	154 ± 19	<0.1
60 d/20 °C	6.3 ± 0.2	6.3 ± 0.3	4.1 ± 0.1	176 ± 3	<0.1
10 d/40 °C	6.7 ± 0.5	7.1 ± 0.6	5.1 ± 0.4	220 ± 19	<0.1
	**Relative Migration [%] into Olive Oil**			**Migration of Styrene [µg/kg]**	**Weight Increase [%] ^1^**
	**Toluene**	**Chlorobenzene**	**Styrene**		
15 d/20 °C	2.9 ± 0.2	2.4 ± 0.1	0.9 ± 0.3	55 ± 14	0.1
20 d/20 °C	3.2 ± 0.1	3.1 ± 0.2	1.0 ± 0.1	55 ± 5	0.2
30 d/20 °C	3.7 ± 0.2	3.6 ± 0.5	1.4 ± 0.3	63 ± 13	0.3
40 d/20 °C	4.5 ± 0.3	4.3 ± 0.4	2.2 ± 0.2	97 ± 9	0.3
60 d/20 °C	5.0 ± 0.2	5.0 ± 0.2	2.4 ± 03	109 ± 15	0.2
10 d/40 °C	7.0 ± 0.8	7.1 ± 0.7	4.8 ± 0.5	211 ± 25	0.2

^1^ since the GPPS material after contact with cream and olive oil broke into pieces, it was not possible to determine precise weight.

**Table 3 molecules-27-00823-t003:** Relative migration [%] of volatiles, migrated amount of styrene in real foods and weight increase [%] of the polymer at the tested time/temperature conditions from HIPS.

Time/Temperature	Relative Migration [%] into Milk (3.5% Fat)			Migration of Styrene [µg/kg]	Weight Increase [%]
	Toluene	Chlorobenzene	Styrene		
15 d/20 °C	1.7 ± 0.3	2.8 ± 0.2	2.2 ± 0.2	91 ± 7	0.1
20 d/20 °C	1.6 ± 0.1	2.5 ± 0.1	1.8 ± 0.1	79 ± 6	0.2
30 d/20 °C	2.5 ± 0.5	3.7 ± 0.6	2.8 ± 0.5	118 ± 16	0.1
40 d/20 °C	2.5 ± 0.4	3.7 ± 0.4	2.6 ± 0.3	111 ± 14	0.2
60 d/20 °C	3.3 ± 0.1	4.8 ± 0.1	3.4 ± 0.1	145 ± 5	0.1
10 d/0 °C	3.4 ± 0.2	4.6 ± 0.3	3.6 ± 0.3	152 ± 13	0.1
	**Relative Migration [%] into Cream (30% Fat)**			**Migration of Styrene [µg/kg]**	**Weight Increase [%]**
	**Toluene**	**Chlorobenzene**	**Styrene**		
15 d/20 °C	3.3 ± 0.2	3.6 ± 0.2	3.6 ± 0.4	149 ± 16	0.5
20 d/20 °C	5.8 ± 1.1	5.9 ± 0.8	5.2 ± 0.7	221 ± 32	0.4
30 d/20 °C	4.5 ± 0.3	4.8 ± 0.1	4.4 ± 0.3	185 ± 16	0.4
40 d/20 °C	5.8 ± 0.8	6.5 ± 0.7	5.5 ± 0.5	233 ± 20	0.6
60 d/20 °C	6.2 ± 0.4	7.2 ± 0.5	5.8 ± 0.4	241 ± 20	0.6
10 d/40 °C	7.0 ± 0.6	8.0 ± 0.2	6.6 ± 0.3	278 ± 10	0.4
	**Relative Migration [%] into Olive Oil**			**Migration of Styrene [µg/kg]**	**Weight Increase [%]**
	**Toluene**	**Chlorobenzene**	**Styrene**		
15 d/20 °C	3.1 ± 0.2	3.3 ± 0.3	2.7 ± 0.4	115 ± 18	0.3
20 d/20 °C	3.4 ± 0.3	3.7 ± 0.4	2.9 ± 0.3	122 ± 14	0.3
30 d/20 °C	4.1 ± 0.3	4.3 ± 0.5	3.2 ± 0.2	137 ± 9	0.2
40 d/20 °C	4.1 ± 0.1	4.4 ± 0.2	3.3 ± 0.2	139 ± 5	0.5
60 d/20 °C	5.9 ± 1.1	6.5 ± 0.9	5.0 ± 1.0	213 ± 41	2.3
10 d/40 °C	5.6 ± 0.4	6.9 ± 0.5	5.3 ± 0.6	222 ± 25	0.3

**Table 4 molecules-27-00823-t004:** Weight increase [%] of GPPS and HIPS after contact with real foods at 5 °C and 20 °C up to storage for 50 or 30 days.

Food/Storage Temperature	Weight Increase [%]						
	GPPS				HIPS		
	10 Days	20 Days	30 Days	50 Days	10 Days	20 Days	30 Days
Lard (100% fat)/5 °C	<0.1	0.1	0.1	0.2	0.5	0.5	0.7
Butter/5 °C	<0.1	0.2	1.1	1.2	0.5	0.6	0.7
Fish oil ^1^/20 °C	<0.1	0.2	<0.1	<0.1	0.5	0.5	0.9
Miglyol^®^ 812 ^2^/20 °C	0.1	0.4 ^4^	0.4 ^4^	0.4 ^4^	3.6	6.1	8.0
Clear orange juice/20 °C	0.1	0.1	<0.1	<0.1	0.2	0.2	0.2
Ground coffee beans/20 °C	<0.1	<0.1	<0.1	<0.1	0.1	0.3	0.4
Noodles ^3^/20 °C	<0.1	<0.1	<0.1	<0.1	<0.1	0.1	<0.1
Oat flakes/20 °C	<0.1	<0.1	<0.1	<0.1	<0.1	<0.1	<0.1
Wheat loops (sugared)/20 °C	<0.1	0.1	<0.1	0.1	0.1	<0.1	0.1
Water/20 °C	<0.1	<0.1	<0.1	<0.1	<0.1	<0.1	<0.1

^1^ Omega-3 total by Norsan (contains natural fish oil, olive oil, mixed tocopheroles, cholecalciferol, natural lemon oil). ^2^ CAS-No. 73398-61-5: Decanoyl- and Octanoylglycerides. ^3^ Made from wheat semolina with 10% egg, noodles crushed to have intimate contact. ^4^ since the GPPS material after contact with Miglyol broke into pieces, it was not possible to determine precise weight.

**Table 5 molecules-27-00823-t005:** Weight increase [%] of GPPS and HIPS in hot filled aqueous foods up to storage for up to 30 min (during cooling process or at constant temperature).

Food/Temperature	Weight Increase [%]					
	GPPS			HIPS		
	10 min	20 min	30 min	10 min	20 min	30 min
Boiling water (starting 100 °C, end 36 °C)	0.1	0.1	<0.1	0.1	0.2	0.3
Hot water (starting 80 °C, end 36 °C)	<0.1	<0.1	<0.1	0.2	0.2	0.2
Hot water (starting 60 °C, end 36 °C)	<0.1	<0.1	<0.1	<0.1	<0.1	<0.1
Warm water (starting 40 °C, end 29 °C)	<0.1	<0.1	<0.1	<0.1	<0.1	<0.1
Brewed coffee (starting 90 °C, end 28 °C)	0.1	0.1	0.1	0.4	0.6	0.6
Brewed coffee (100 °C constant)	0.3	0.4	0.3	0.4	0.6	1.2

**Table 6 molecules-27-00823-t006:** Migration of styrene in food simulants and real foods at different time/temperature testing conditions, weight increase and visual changes of the sample material.

Food (Simulant) Time-Temperature	GPPS			HIPS		
	Styrene Migration ^a^ [µg/kg]	Weight Increase [%]	Visual Changes ^b^	Styrene Migration ^a^ [µg/kg]	Weight Increase [%]	Visual Changes ^b^
Isooctane@10 d/20 °C ^c^	39	<0.1	no	2665	3	yes
Isooctane@10 d/40 °C ^c^	126	0.2	no	3492	12	yes
Isooctane@10 d/60 °C ^c^	4248	45	yes	3425	30	yes
Isooctane@1 d/20 °C ^d^	23	<0.1	no	1788	1.6	yes
Isooctane@1 d/40 °C ^d^	50	0.1	no	3156	5.6	yes
Isooctane@1 d/60 °C ^d^	1423	12	yes	3366	21	yes
95% ethanol@10 d/20 °C ^c^	<20	1.7	no	190	2.7	no
95% ethanol@10 d/40 °C ^c^	690	2.6	no	970	2.7	no
95% ethanol@10 d/60 °C ^c^	3329	4.1	yes	3617	3.2	yes
95% ethanol@1 d/40 °C ^d^	239	1.3	no	227	1.6	no
95% ethanol@1 d/60 °C ^d^	833	3	yes	980	3.2	yes
50% ethanol@10 d/20 °C ^c^	<40	0.8	no	48	0.9	no
50% ethanol@10 d/40 °C ^c^	<40	1.1	no	297	1.3	no
50% ethanol@10 d/60 °C ^c^	598	1.2	no	341	1.3	yes
Milk@10 d/40 °C	92	0.1	no	152	0.1	no
Cream@10 d/40 °C	220 ^e^	<0.1 ^e^	no	278	0.4	no
Olive oil@10 d/40 °C	211 ^e^	0.2 ^e^	yes	222	0.3	no

^a^ applying a surface/volume ratio of 6 dm^2^/kg (according to the EU cube model). ^b^ all the photos of the sample materials are given in the Appendix A. ^c^ official condition according Regulation (EU) 10/2011. ^d^ alternative condition according [8]. ^e^ since the GPPS material after contact with cream and olive oil broke into pieces, it was not possible to determine precise migration and weight increase.

**Table 7 molecules-27-00823-t007:** Within laboratory detection limit in food simulants [µg/mL] and real foods [µg/g].

Substance	Limit of Detection in Food Simulants [µg/mL] and Real Foods [µg/g]					
	95% Ethanol	50% Ethanol	Isooctane	Olive Oil	Cream	Milk
Toluene	0.58	1.95	1.29	0.06	0.02	0.01
Chlorobenzene	0.57	1.64	0.23	0.04	0.04	0.01
Styrene	0.22	1.43	4.40	0.03	0.06	0.01
Phenyl cyclohexane	0.60	0.74	2.14	n.d.	n.d.	n.d.
Benzophenone	0.20	0.18	1.96	n.d.	n.d.	n.d.
Methyl stearate	2.16	7.50	1.60	n.d.	n.d.	n.d.

n.d. not determined.

## Data Availability

Not applicable.

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
