# Peer review of "Migration Testing of GPPS and HIPS Polymers: Swelling Effect Caused by Food Simulants Compared to Real Foods"

_molecules, 2022, doi:10.3390/molecules27030823_

Round 1
Reviewer 1 Report
The authors evaluated the migration of GPPS and HIPS polymers to food simulants and real foods (milk, cream and olive oil) at different time/temperature conditions, showing kinetic migration data for several model substances, styrene and styrene oligomers. This study compared migration values and also swelling effect and visual changes caused by the different food media and conditions. The work has novelty and scientific significance. This manuscript is appropriate for "Molecules" and some improvements are should be done for publication.
Abstract:
It should be improved highlighting the most important results found through the comparative study between GPPS and HIPS, food simulants and real food, time/temperature conditions.
Intro:
-It should be included references/previous works (if it is possible) on migration studies of polymers different than PS using food simulants and real foods to broaden the context (agreement with the Regulation EU N° 10/2011).
-Line 46: Briefly explain the important features and formation process of expanded PS as it was done for GPPS and HIPS, and quantify their uses in the food packaging sector.
-Line 95-98: ¿What does the following sentence mean? "...the styrene monomer concentrations in foods have not significantly changed since the 1980s"...the idea is not clear. Please, clarify the explanation.
-Line 122-125: It is not clear how many and which food simulants and food types were used. Please, extend the explanation.
Results and discussion:
Table 1: Please, include in the legend of the table how many measurements for each experiment were done to calculate the standard deviation.
-Line 179: ¿The values higher than 100% could be associated with the heterogeneity of the concentration (Cp,o) along with the sheet? ¿The explanation could be supported with the Cp,o + SD values of Table 1?
-Figure 2a: The Y-axis scale (Relative migration) should be modified for a better reading of the data. Most appropriate scale: 0,2,4...
-Delve into the explanation of some findings, for example, ¿Which could be the reason GPPS was swollen more than HIPS in isooctane at 60 °C but the contrary occurred at 40 °C?
-Figure 9: The legend of the figure is on the X-axis title. Please, move it.
-The thickness and color tone of the lines of the graph area should be unified. They appear to have different shades of gray. It could be suitable to include borderlines in the graph area of all figures (a, b, c and d).
Materials and Methods:
-It should be interesting to include the molecular weights of the used PS. The sheets were prepared by cast extrusion and which extrusion conditions were used?
-Line 694: "A subsequent extraction and recovery experiments were performed to check for exhaustiveness". Please, give details about the conditions of the extraction and recovery processes.
-Line 739-740: Please, explain better or standardize the units of the surface-to-volume ratio used with the food simulants (2 dm2/100 mL) and real foods (0.2 dm2/20 g) to clarify the equivalence for comparative effects.
-Line 765: Please, clarify why the total contact area was 54 cm2.
Conclusions:
Please, conclude about the possible factors or phenomena that could influence the mean differences observed of the results between GPPS and HIPS from the point of view of structural changes with the condition (time/temperature/food type or simulant). Explanations also can be included in the corresponding discussion section if necessary.
Reviewer 2 Report
This manuscript closely follows the research trends of toxicology and FCM oral exposure risk assessment, considering the physicochemical properties of styrenic polymers, series of the relative migration of the spiked substances (volatile and low-volatile) as well as styrene monomer and styrene oligomer, weight increase and visual changes were conducted with both food simulant and real food at different conditions. The impact factors of migration behavior which affected by swelling effect has been studied since dairy products, fatty food simulants and alternative simulants specified in Annex III of EU Regulation 10/2011 may cause swelling at certain temperatures and times, especially in 95% ethanol and isooctane which may lead to non-compliance because the overestimate of migration compared with real food. The topic is meaningful, clear thinking, the data are detailed and the conclusions are reasonable. It gives suggestions for improving the migration test conditions of the material and provides references for migration limits and oral exposure risk assessment.
Here are some questions should be checked and modified.
- Some normal information of the HIPS and GPPS is needed.
- Why chose the food simulants 20% ethanol, 10% ethanol and 3% acetic to observe weight increase in chapters 2.4.2 ? If the purpose is to compare with the real food below, why does the processing time differ so much and how did the authors consider the different real food storage times?
- Line 344: In ‘at the same condition it reached 20% in case of HIPS.’ Please check ‘20%’which is’15%’ in Figure 8.
- Line 362: ‘The relative migration of styrene was 10% after 10 days and 5% for the other volatiles.’, the data are inconsistent with the information in Figure 9.
- Line 443: ‘In olive oil and milk, the migration was twice as high compared to 40 ℃’ Please check if 40℃ should be 20℃.
- Figure 5. Missed C and D.
Reviewer 3 Report
The authors of this paper have addressed the issue of kinetic migration from polystyrene (GPPS) and high impact polystyrene (HIPS) for a set of food imitation model substances, which is very important in terms of food security of citizens. The degree of swelling of the polymer was characterized gravimetrically and visual changes of the studied samples after migration contact were recorded. Visual changes and a strong swelling effect caused by these simulants were observed especially at high temperatures. Substances that can migrate from polystyrene (PS) into foods and beverages are the remaining monomers (i. e. , styrene and butadiene), oligomers (mainly styrene dimers and trimmers), and any additives used (e. g. , antioxidants, antioxidant agents, and extenders). The study of styrene migration was thoroughly investigated, strongly dependent
on its residue level in the polymer, the fat content of the food, as well as the storage temperature and time conditions.
The authors obtained kinetic migration data for model substances as well as for styrene monomer and styrene oligomers under different time-temperature conditions using 50% ethanol, isooctane and 95% ethanol as food simulants. The migration of volatile compounds was quantified in UHT milk (3. 5% fat), UHT cream (30% fat) and olive oil.
General comments on the article:
Title:
The title of the article reflects the content of the conducted research
Synopsis:
In my opinion the most important results should be quoted and general information can be omitted
Introduction
The chapter is well written, addressing the major issues that were addressed in the paper.
Results
The main results are presented as graphs reflecting the kinetics of styrene migration using food simulants. This way of presenting results is not objectionable, it is clear and easy to interpret. The tables are clear and well structured. The description of the results is very accurately described and not objectionable.
Discussion
The chapter was correctly written, it mainly referred to the kinetic migration from polystyrene (GPPS) and high impact polystyrene (HIPS) for a set of polystyrene food simulants. The chapter discusses the results obtained and refers to the results of other authors
Material and methods
The analytical methods are correctly described, without being objectionable
References:
The amount and selection of literature is appropriate.
I believe that this article is suitable for publication in the journal Molecules after completing the abstract.
Round 2
Reviewer 2 Report
This version can be accepted and published.